# Morphological and Molecular Investigations of Aquaporin-7 (AQP-7) in Male *Camelus dromedarius* Reproductive Organs

**DOI:** 10.3390/ani13071158

**Published:** 2023-03-25

**Authors:** Thnaian A. Al-Thnaian

**Affiliations:** Department of Anatomy, College of Veterinary Medicine, King Faisal University, P.O. Box 400, Al-Hassa 31982, Saudi Arabia; talthnaian@kfu.edu.sa; Tel.: +966-558122112

**Keywords:** aquaporin-7, *Camelus dromedarius*, epididymis, prostate, testes

## Abstract

**Simple Summary:**

This study aimed to investigate the immune reactivity levels of aquaporin-7 (AQP-7) antibody in the male *Camelus dromedarius* testis, epididymis, ductus deferens and prostate gland. This technique is potentially useful for providing insight into the predicted role of AQP-7 during rutting and non-rutting seasons in male camels. Therefore, it is important to check the immune reactivity levels of the anti-AQP-7 antibody as a marker for water and energy homeostasis in the reproductive tract of male camels during rutting and non-rutting seasons.

**Abstract:**

Aquaporins (AQP) are involved in bidirectional transfers of water and small solutes across cell membranes. They are present in all tissues. However, the expression of AQP-7 has not yet been demonstrated in the reproductive tract of the camelid *Camelus dromedarius*. The study presented here concerns the immunohistochemical evidence of aquaporin-7 (AQP-7) in different parts of the male genital tract of *Camelus dromedarius***.** To check the immune reactivity levels of anti-AQP-7 antibody in the male genital tract of *Camelus dromedarius*, the testes (proximal part, distal part and rete testis), epididymis (head, body and tail), ductus deferens (initial, middle and ampullary part) and prostate gland (compact and disseminated part) were collected from 12 male camels during the rutting and non-rutting seasons and subjected to immunohistochemistry. The result showed that the highest level of AQP-7 mRNA expression was in the testis of rutting and non-rutting males compared to the ductus deferens, epididymis and prostate. In addition, the highest mRNA gene expression of AQP-7 was in rutting males compared to non-rutting males. AQP-7 mRNA expression was higher in the ret testis, the body of the epididymis, the ampullary part of the ductus deferens and the compact part of the prostate. The immune reactivity levels of AQP-7 in rutting males showed strong reactivity in the testis and prostate compared to the epididymis and ductus deferens. On the basis of the results, it can be concluded that the distribution of the AQP-7 transcript and protein varied among rutting and non-rutting seasons and that the physiological roles of AQP-7 in the transportation of lipids, energy and water should be considered the main challenge in the activity and establishment of male *Camelus dromedarius* fertility during the rutting and non-rutting seasons. Moreover, AQP-7 detection is critical in assessing regulation and screening for new modulators that can prompt the development of effective medication to enhance fertility during rutting and non-rutting seasons.

## 1. Introduction

Aquaporins (AQPs) are expressed in almost every organism and tissue and are involved in the bidirectional transfer of water and small solutes across cell membranes in response to osmotic and hydrostatic pressure or concentration gradients [1]. Mammalian AQPs (AQP0-12) are grouped according to their primary structure [2] and permeability [3] into (1) orthodox aquaporins (AQP0, AQP1, AQP2, AQP4, AQP5, AQP6 and AQP8) that are considered primary selective to water; (2) aquaglyceroporins (AQP3, AQP7, AQP9 and AQP10) that transport water as well as glycerol and other small neutral solutes; and (3) subcellular or unorthodox aquaporins (AQP11 and AQP12) with distinct evolutionary pathways, which are localized in intracellular membranes [1,4,5,6]. AQP7 is considered to constitute a crucial for body lipid and energy homeostasis [7]. Due to their selective permeability, aquaglyceroporins have important roles in glycerol accumulation and metabolism in many tissues such as the testes [8], epididymis [9], and ductus deferens [10]. AQP7 is also expressed in sperm [11]. Aquaglyceroporins (AQP7 and AQP9) have also been investigated in efferent ducts and the epididymis in dogs by examining their collaborative role and hypothesizing their changed activities according to different stimuli (as a cause of cryptorchid condition) [12].

The expression of AQP-7 in the male *camelus dromedaries* reproductive tract has not been reported. The purpose of the present study was to examine the expression of AQP-7 at the level of mRNA and protein in the testis, epididymis, ductus deferens and prostate. Therefore, this study proposed that the expression of AQP-7 may be differently expressed in various segments of the male reproductive tract according to its contribution to form a specific fluid environment in the reproductive tract.

## 2. Materials and Methods

### 2.1. Pre-Experimental Ethics and Experimental Sampling

All animal sampling procedures were performed according to the animal slaughter protocols and ethical guidelines of the Ministry of Municipal, Rural Affairs, and Housing, Saudi Arabia. The King Faisal University Ethics Committee approved the animal sampling (Ref. No. KFU-REC-2023-MAR-ETHICS718). The tissue samples, as shown in Table 1, were collected from adult dromedary camel bulls from the Al-Omran slaughterhouse with an age range of 4 to16 years.

### 2.2. Immunohistochemistry Paraffin Protocol (IHC-P) for AQP-7 Detection

Approximately one centimeter of each tissue type mentioned in Table 1 was collected from 12 male camels during rutting and non-rutting seasons and then fixed overnight in 4% paraformaldehyde. All the fixed samples were processed for immunohistochemical examination according to the following steps: dehydration, clearing, infiltration, embedding and sectioning. Tissue sections were then deparaffinized by passing through the thermo-scientific super-frosted plus charged slides (Thermo Scientific, Waltham, MA, USA, 4951PLUS-001) in xylene with two changes for 5 min each. This was followed by hydration of the sections by dipping them for 30 s in degraded alcohol (100%, 100%, 95%, 80%, 70% and washed the slides 2 × 5 min in tris-buffered saline (1X TBS) plus 0.025% triton X100 with gentle agitation; this was followed by blocking in 10% normal serum with 1% bovine serum albumin (BSA) in (1X TBS) for 2 h at room temperature. A mouse- and rabbit-specific HRP/DAB (ABC) detection IHC kit (ab64264) was used to detect anti-AQP-7 antibodies (Abcam, Cambridge, Cambridgeshire, UK, Ab15123) as follows: enough drops of hydrogen peroxide block were added to cover the sections which were incubated for 10 min at room temperature and then washed twice in 1X TBS buffer; protein block was applied on sections and which were incubated for 10 min at room temperature and then washed twice in 1X TBS buffer; the primary mouse anti-AQP-7 antibody (Abcam, Ab15123) was diluted in 1X TBS with 1% BSA and applied on sections and these were incubated overnight at 4 °C.

The next day, the sections were rinsed four times in 1X TBS 0.025% triton with gentle agitation, and biotinylated goat anti-polyvalant enzyme-conjugated secondary antibody was applied to the sections, which were incubated for 1 h at room temperature and washed four times in 1X TBS 0.025% triton. Streptavidin peroxidase was applied to cover the sections, which were incubated for 10 min at room temperature and rinsed four times in 1X TBS 0.025% triton. A measure of 30μL DAB substrate solution was then applied to cover the sections until the desired color intensity was reached, and then the samples were washed with PBS three times for 2 min each. Then, the sections were dehydrated by dipping them in graded alcohol (70%, 95%, 100%, and 100%) for a few seconds and cleared using xylene with two changes, for 5 min each. Finally, the sectioned tissues were mounted by using DPX with cover slides measuring 22 × 40 mm, and the staining was observed by Leica ICC50 W light microscopy under 10× and 40× magnification powers, with a Wi-Fi-capable digital camera detector and Leica AirLab App software. The reaction reactivity was assessed using image processing and analyzed using an ImageJ 1.52a analyzer (Wayne Rasband, National Institute of Health, Bethesda, MD, USA, http://imagej.nih.gov/ij) “URL (accessed on 15 September 2020)”.

### 2.3. RNA Isolation and Semi-Quantitative Real Time RT-PCR Analysis

AQP-7 mRNA expression levels were evaluated using a semi-quantitative real-time RT-PCR analysis. Tissues were homogenized by Bead Ruptor (24 Bead Mill Homogenizer, OMNI, USA) and total RNA was extracted using the PureZOL TM RNA isolation method (BIO-RAD, Catalog #732-6890, Hercules, CA, USA) by adding 1 mL of PureZOL for every 100 mg of tissue in a 2 mL-sized tube for disruption and homogenization by Bead Ruptor (24 Bead Mill Homogenizer OMNI, USA). Then, 0.2 mL of chloroform was added per 1 mL of PureZOL, followed by incubation for 5 min at room temperature and centrifugation at 12,000× *g* for 15 min at 4 °C. The aqueous phase was immediately transferred to a new RNase-free tube. A measure of 0.5 mL of isopropyl alcohol was used per 1 mL of PureZOL, followed by incubation at room temperature for 5 min and then centrifugation at 12,000× *g* for 10 min at 4 °C. The RNA pellet was washed by adding 1 mL of 75% ethanol and was then centrifuged at 7500× *g* for 5 min at 4 °C. RNA pellet was allowed to air-dry for about 5 min and then resuspend in 100 µL of RNasefree water. DNA was removed using a DNase I kit (Ambion, AM2222, Carlsbad, CA, USA), and the RNA samples were checked for their concentration and purity (260:280 nm absorbency) using a Synergy™ Mx Monochromator-Based Multi-Mode Microplate Reader (Bio-Tek, Winooski, VT, USA). RNA (2 μg) was reverse transcribed to cDNA in a reaction mixture using an iScriptc DNA synthesis kit (BIO-RAD, Catalog #170-8890, Hercules, CA, USA).

A semi-quantitative real-time CFX96 TouchTM Real-Time PCR analysis (BIO-RAD, Hercules, CA, USA) was performed using the ssoAdvancedTM SYBR Green Supermix kit (BIO-RAD, Catalog #170-5270, Hercules, CA, USA). The 20 µL reaction mix was prepared from 10 µL of the master mix, 2 µL of the forward primer pm/μL, 2 µL of the reverse primer pm/μL (Table 2), 2 µL of cDNA from the sample, and 4 µL of nuclease-free water. The cycling parameters were 95 °C for 1 min, 40 cycles at 95 °C for 10 s, followed by 30 s at 60 °C and 72 °C for 10 s, with a final melting at 95 °C for 20 s. Triplicates from each cDNA were analyzed, fluorescence emission was detected, and relative quantification was calculated automatically according to the GAPDH housekeeping gene.

### 2.4. Statistical Analysis

The data for AQP-7 protein and gene expressions were expressed as means ± SE. A two-way ANOVA and an all-pairs Bonferroni test were applied to compare the different parameters in each treatment group using IBM SPSS Statistics 20 software (IBM, Armonk, NY, USA). Differences were considered significant at *p* < 0.05.

## 3. Results

### 3.1. AQP-7 Gene Expression

The gene expression of AQP-7 is shown in Figure 1; a higher gene expression of AQP-7 was in observed in the testis compared to the epididymis, ductus deferens and prostate. In the testis, mRNA gene expression of AQP-7 was varied; the highest gene expression was found in RT compared to TP and TD in both the rutting and non-rutting season, as shown in Figure 1. Moreover, mRNA expression of AQP-7 was significantly higher in the rete testis of rutting males compared to non-rutting males.

In the epididymis, the highest gene expression of AQP-7 was in the EB compared to EH and ET in both rutting and non-rutting males. The gene expression of AQP-7 was significantly higher in all parts of the epididymis in rutting males compared to non-rutting males. In the ductus deferens, a higher value of gene expression of AQP-7 was detected in DA in both rutting and non-rutting males compared to the DI and DM. The highest gene expression was highest in the ductus deferens of rutting males compared to the ductus deferens of non-rutting males. Furthermore, the PD in both rutting and non-rutting males also had a higher gene expression of AQP-7 compared to the PC of both rutting and non-rutting males, as shown in Figure 1.

### 3.2. Immunohistochemistry of AQP-7

#### 3.2.1. Testes

The immune reactivity levels of anti-AQP-7 antibody are shown in Table 3. In the rutting season, the reactivity was intense in the TP, TR and TD (Figure 2). In non-rutting male camel testes, the reactivity was intense in the TR and TD of the testes compared to the TP, as shown in Table 3. The immunohistochemical detection of AQP-7, localized in the seminiferous tubules in the TP and TD of the testes and in the interconnecting tubules in TR, are shown in Figure 2.

#### 3.2.2. Epididymis

In the rutting season, the immune reactivity levels of anti-AQP-7 antibody were intense in the head, body and tail compared to the weak reactivity of the same tissues of non-rutting males. The signal was strong in the basal cell (stem cell) layer and sterocelia of the columnar cell of the head, body and tail of the epididymis (Figure 3). In the non-rutting male camel epididymis, the immune reactivity of the anti-AQP-7 antibody was very weak in the head and body and weak in the tail, as shown in Table 3. The signal shows a very weak positivity in both the basal cell and columnar cell (Figure 3).

#### 3.2.3. Ductus Deferens

In rutting males, intense immune reactivity of anti-AQP-7 antibody was found in the DI, DM and DA of ductus deferens and weak reactivity was found in the DI and DM, whereas moderate reactivity was observed in the DA of non-rutting males, as shown in Table 3. The immune reactivity of anti-AQP-7 in ductus deferens of rutting males shows an intense positivity in the epithelium and basal cell of the initial and middle part of the ductus deferens, compared to the weak signal in both the DI and DM of the ductus deferens of non-rutting males (Figure 4). The DA of the ductus deferens of rutting males shows strong positivity in the epithelial columnar and basal cell compared to the moderate reactivity of non-rutting males (Figure 4).

#### 3.2.4. Prostate Gland

In the rutting season, the immune reactivity of anti-AQP-7 antibody was intense in both PC and PD, as shown in Table 3. The signal shows strong positivity in the acinar cells of the PC and an intense signal in the acinar cells and secretory alveoli of the PD (Figure 5). The immune reactivity of anti-AQP-7 antibody in the non-rutting season was intense in the PC compared to the moderate reaction seen in the PD, as shown in Table 3. The signal shows strong positivity in the acinar cells of the PC and moderate positivity in the acinar cell and secretory alveoli of PD (Figure 5).

## 4. Discussion

The male camel is described as a seasonal breeder with a marked peak in sexual activity (the rut) during the breeding season, and it is generally thought that the male is sexually quiescent for the remainder of the year but remains capable of mating with and fertilizing an estrous female at any time of the year [13]. In the literature, information on the stimuli for the onset of the dromedary camel breeding season is rather conflicting. Some studies showed that decreasing daylight appeared to be the stimulus to seasonality [14,15,16]. Globally, the breeding season of the camels begins at different dates, beginning in September and ending at different dates until June in different parts of the northern hemisphere and ranging from June to September in the southern parts of the world, which are the mildest periods of the year and have decreasing and/or increasing daylight, while the non-breeding season is in hot summer months [13,17]. However, the data on the breeding season in the dromedary are rather conflicting. In Saudi Arabia, the breeding season has been reported from October to April [17,18]. In rutting, the seminiferous tubules have a greater diameter (209–220 μ) than in the tubules of camels not in rut (190–203 μ); spermatogonia, spermatids and spermatozoa also become numerous [13]. The activity of the Leydig cells becomes maximal during the rutting season, but they are less active in the non-breeding season, with a resulting reduction in steroidogenic activity by the testes [19].

Our results show that the mRNA expression of AQP-7 was significantly higher in the testes of rutting males compared to testes of non-rutting males. Moreover, mRNA gene expression of AQP-7 in testicular parts was varied in both the rutting and non-rutting season; the highest gene expression was found in the rete testis compared to the proximal and distal parts in both the rutting and non-rutting season. This may be supported by the finding of Marai et al. [13] (Marai et al., 2009), in that the seminiferous tubules have a greater diameter during the rutting season compared with the non-rutting season.

There are two diffusion pathways through which water crosses cell membranes: diffusion of water molecules across the hydrophobic bilayer and diffusion through specialized protein channels known as aquaporins (AQPs) [20]. Ishibashi, Tanaka [6] reported that AQPs can be divided into three subgroups: the classical AQPs (water-selective), aquaglyceroporins (glycerol, water and small-solute permeable) and S-aquaporins (water-permeable). Due to their unique ability to transport glycerol, AQPs play critical roles in osmoregulation by controlling the intracellular accumulation of glycerol at the level of gene expression, metabolism and transport [21].

Water homeostasis and energy balance are critical for reproductive cell survival during the rutting and non-rutting season and the process of water movement in the excurrent duct system of the male reproductive tract is pivotal for the establishment of male fertility [10,22]. In the testis, fluid is continuously produced by the seminiferous epithelium and aids in transporting the sperm toward the rete testis [10]. Our data show that the reactivity of anti-AQP-7 antibody was intense in the testis of the male camel during the rutting season compared to non-rutting males; this may indicate that the seminiferous tubules during the rutting season were more active and that AQP-7 may play a significant role in water homeostasis and a pivotal role in the establishment of male fertility.

Following the completion of spermatogenesis, sperm leaves the testis through the efferent ducts and enters the epididymis. Although much of the fluid leaving the testicle is resorbed within the efferent ducts, water continues to be removed from the lumen of the epididymis [23]. Our findings showed that the highest gene expression of AQP-7 was in the body of epididymis compared to the head and tail in both rutting and non-rutting males. In addition, the immune reactivity levels of anti-AQP-7 antibody were intense in the head, body and tail compared to the weak reactivity in the same tissues of non-rutting males. The signal was strong in the basal cell (stem cell) layer and sterocelia of the columnar cell of the head, body and tail of the epididymis. This was compatible with the findings of Squillacioti et al., which showed regional tissue distributions of AQP7, particularly at the level of the epithelium of efferent ductules and in both the regions, caput and cauda, of the canine cryptorchid epididymis [12]. In the ductus deferens, the higher value of gene expression of AQP-7 was in the ampullary part in both rutting and non-rutting males compared to the initial and middle parts. The highest gene expression was in the ductus deferens of rutting males compared to the ductus deferens of non-rutting males. This may indicate that AQP-7 may also play a significant role in water homeostasis and the post-testicular maturation of spermatozoa during transit through the epididymis and ductus deferens, including the acquisition of progressive motility and the ability to undergo capacitation [24,25,26].

Prostatic secretion of prostate fluid is one of the important components of semen; it provide nutrients and a suitable environment for sperm. An and Wang [27] reported that AQP1 and AQP2 were localized in the prostate, showing that the water channel protein is closely related to the function and may influenced the transportation of sperm and nutrition. Our present finding showed that the immune reactivity of anti-AQP-7 antibody was intense in both compact and disseminated parts of the prostate gland during the rutting season. Furthermore, the disseminated part of the prostate in both rutting and non-rutting males has a higher gene expression of AQP-7 compared to the compact part of both rutting and non-rutting males. This may indicate that the water channel protein may play an important role in prostatic secretion, the process that provides nutrition for sperm, and may influence the liquid secretion during the transportation of sperm. Furthermore, AQP7 may have a significant role in maintaining water equilibrium during the secretion of seminal and prostatic fluid [28]. In mammals, aquaglyceroporins (AQP-7) are involved in energy metabolism by controlling the glycerol content in the epidermal layer of various tissues and organs, which has a key role in the regulation of metabolic and energy homeostasis [29,30]. Therefore, this study aimed to investigate and detect AQP-7 activity across the male reproductive tracts of camels during the rutting and non-rutting season. In addition, this experiment is required to cover the level of aquaglyceroporin (AQP-7) mRNA expression in the male camel during breeding season and quiescent period.

## 5. Conclusions

Based on the results, it can be concluded that the distribution of AQP-7 transcript and protein was varied among rutting and non-rutting seasons and the physiological roles of AQP-7 in the transportation of lipid, energy and water is considered to be a main challenge in the activity and establishment of male fertility during the rutting and non-rutting seasons. Moreover, AQP-7 detection is critical for assessing the regulation of and screening for new modulators that can prompt the development of effective medications to enhance fertility during the rutting and non-rutting seasons.

## Figures and Tables

**Figure 1 animals-13-01158-f001:**
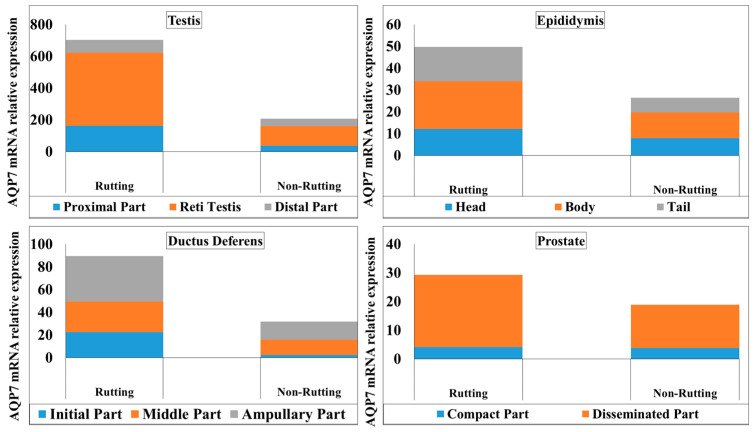
Relative normalized expression of mRNA levels of AQP-7 in testis, epididymis, ductus deferens and prostate.

**Figure 2 animals-13-01158-f002:**
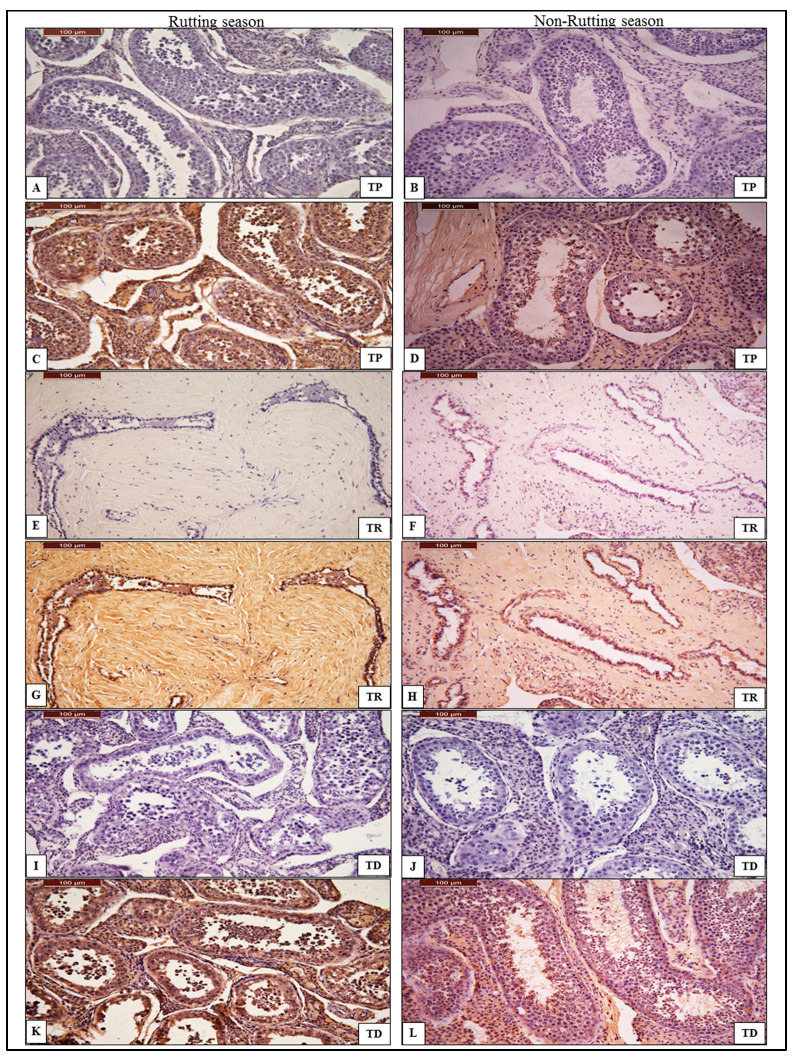
Chromogenic immunohistochemistry of anti-AQP-7 antibody in the proximal part (TP), rete testis (TR) and distal part (TD) of the testes. (**A**,**E**,**I**): negative control of anti-AQP-7 antibody in TP, TR and TD during rutting season. (**C**,**G**,**K**): positive reactions of AQP-7 in TP, TR and TD of the testes show immunohistochemical detection of AQP-7 localized in seminiferous tubules with normal spermatogenesis during rutting season. (**B**,**F**,**J**): negative control of AQP-7 in TP, TR and TD of the testes during non-rutting season. (**D**,**H**,**L**): weak positive reaction of AQP-7 in TP, TR and TD of the testes during non-rutting season shows immunohistochemical detection of AQP-7 localized in the sperm head. AQP-7 is stained by DAB/chromogen and counter-stained with hematoxylin under magnification power 40×.

**Figure 3 animals-13-01158-f003:**
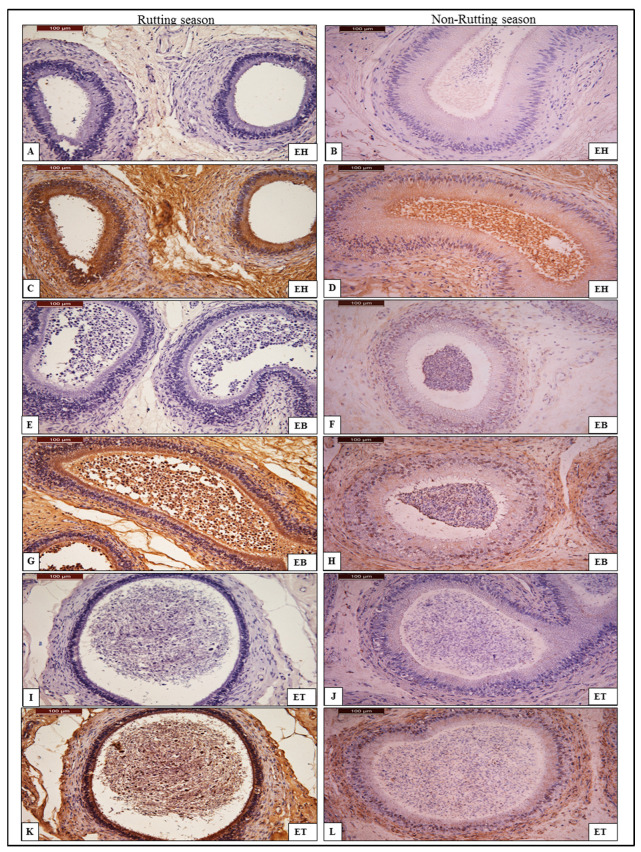
Chromogenic immunohistochemistry of anti-AQP-7 antibody in head (EH), body (EB) and tail (ET) of epididymis. (**A**,**E**,**I**): negative control of anti-AQP-7 antibody in EH, EB and ET during rutting season; (**C**) intensely positive reaction of AQP-7 in EH during rutting season shows strong positivity in basal cell (stem cell) layer and sterocelia of the columnar cell; (**G**) positive reaction of AQP-7 in EB during rutting season shows intense positivity of sterocelia of the columnar cell and (**K**) positive reaction of AQP-7 in ET during rutting season shows intense positivity of columnar cell. (**B**,**F**,**J**): negative control of AQP-7 in EH, EB and ET during non-rutting season; (**D**,**H**,**L**) are positive reactions of AQP-7 in EH, EB and ET during non-rutting season, which shows very weak positivity in both the basal cell and columnar cell. AQP-7 is stained by DAB/chromogen and counter-stained with hematoxylin under magnification power 40×.

**Figure 4 animals-13-01158-f004:**
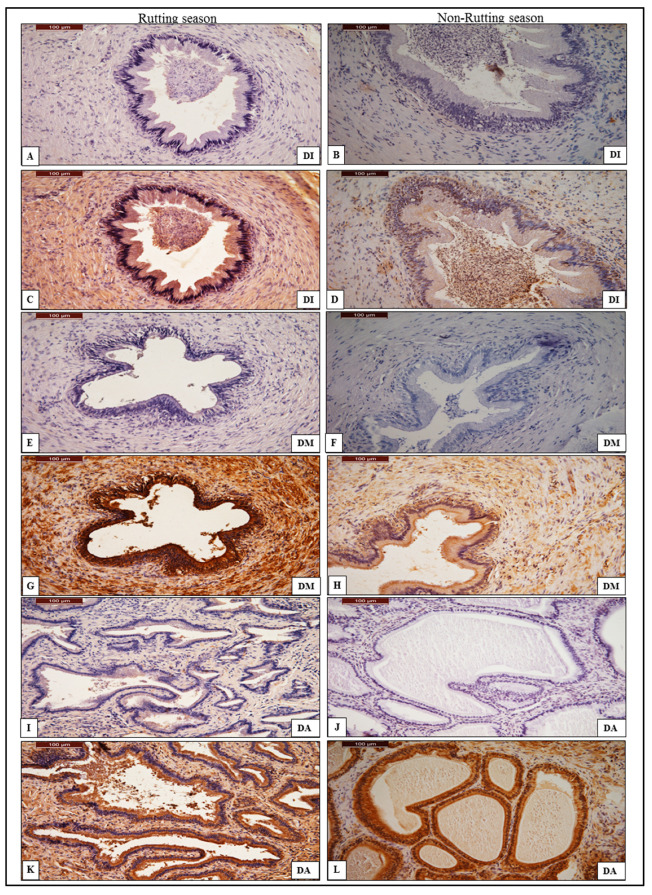
Chromogenic immunohistochemistry of anti-AQP-7 antibody in the initial part (DI), middle part (DM) and ampullary part of ductus deferens (DA). (**A**,**E**,**I**): negative control of anti-AQP-7 antibody in DI, DM and DA during rutting season; (**C**,**G**,**K**): positive reaction of AQP-7 in DI, DM and DA of ductus deferens during rutting season shows very intense positivity of AQP-7. (**B**,**F**,**J**): negative control of AQP-7 in DI, DM and DA of ductus deferens during non-rutting season; (**D**,**H**): positive reaction of AQP-7 in DI and DM of ductus deferens during non-rutting season shows weak positivity of AQP-7. (**L**): positive reaction of AQP-7 in the ampullary part of ductus deferens during non-rutting season shows moderate positivity of the columnar cell. AQP-7 was stained by DAB/chromogen and counter-stained with hematoxylin under magnification power 40×.

**Figure 5 animals-13-01158-f005:**
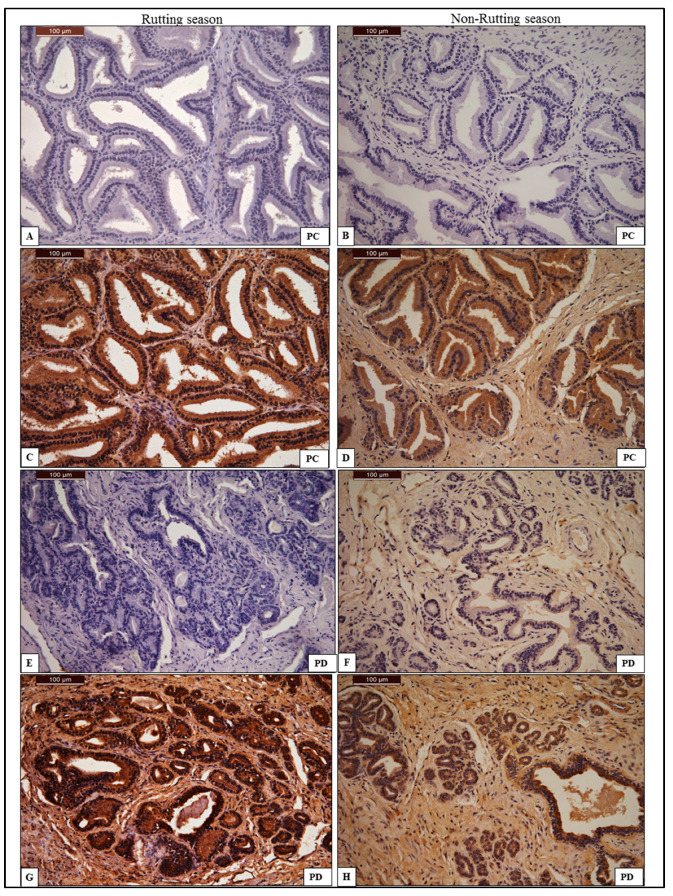
Chromogenic immunohistochemistry of anti-AQP-7 antibody in the compact part (PC) and disseminated part (PD) of the prostate gland. (**A**,**E**): negative control of anti-AQP-7 antibody in PC and PD during rutting season; (**C**) positive reaction of AQP-7 in the compact part of the prostate gland during rutting season shows strong acinar cell positivity of anti-AQP-7; (**G**) positive reaction of AQP-7 in the disseminated part of the prostate gland during rutting season shows strong acinar cell and secretory alveoli positivity of anti-AQP-7. (**B**,**F**): negative control of anti-AQP-7 antibody in PC and PD during non-rutting season; (**D**,**H**) positive reaction of AQP-7 in the PC and PD of the prostate gland during non-rutting season shows strong acinar cell positivity of anti-AQP-7. AQP-7 is stained by DAB/chromogen and counter-stained with hematoxylin under magnification power 40×.

**Table 1 animals-13-01158-t001:** Experimental Sampling of Reproductive Tissues during Rutting and Non-Rutting Season from Adult Dromedary Camel Bulls (*Camelus dromedaries*).

Tissue	Location of Tissue Sample		
Testis	Proximal part (TP)	Distal part (TD)	Rete testis (TR)
Epididymis	Head (EH)	Body (EB)	Tail (ET)
Ductus deferens	Initial part (DI)	Middle part (DM)	Ampullary part (DA)
Prostate	Compact part (PC)		Disseminated part (PD)

**Table 2 animals-13-01158-t002:** Real-time qPCR primer sequences targeting gene expression of AQP-7.

Primers	Sequence	GenBank Reference
AQP-7	F-5-CAGAGAGGAAGAGGCGGTCT-3 R-5-CGTGAACTCGGCTAGGAACT-3	XM_010975828.2
GAPDH	F-5-TCGATCCCCCAACACACTTG-3	XM_010990867.2
	R-5-TGATGGTGCATGACAAGGCA-3	

**Table 3 animals-13-01158-t003:** The semi-quantitatively immunohistochemical reactivity to anti-AQP-7 antibody in testis, epididymis, ductus deferens, prostate and bulbourethral gland.

Immunoreactivity of Rabbit Polyclonal Anti-AQP-7 Antibody											
Tissue Sample	Testis			Epididymis			Ductus Deferens			Prostate Gland	
	TP	TR	TD	EH	EB	ET	DI	DM	DA	PC	PD
Rutting	+ + + +	+ + + +	+ + + +	+ + + +	+ + + +	+ + + +	+ + + +	+ + + +	+ + + +	+ + + +	+ + + +
Non-Rutting	+	+ + + +	+ + + +	+	+	+ +	+ +	+ +	+ + +	+ + + +	+ + + +

(TP): testis proximal part, (TR): rete testis, (TD): testis distal part, (EH): epididymis head, (EB): epididymis body, (ET): epididymis tail, (DI): ductus deferens initial part, (DM): ductus deferens Middle part, (DA): ductus deferens Ampullary part, (PC): prostate compact part and (PD): prostate disseminated part. (+) = very weak reactivity. (++) = weak reactivity, (+++) = moderate reactivity and (++++) = intense reactivity.

## Data Availability

The data presented in this study are available on request from the corresponding author. The data are not publicly available.

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
