# Peer review of "Morphological and Molecular Investigations of Aquaporin-7 (AQP-7) in Male Camelus dromedarius Reproductive Organs"

_animals, 2023, doi:10.3390/ani13071158_

Round 1
Reviewer 1 Report
The manuscript albeit well conceived has several flaws that need to be revised to answer some questions and to improve its quality. Moreover, it needs to be completely revised for the English form because several sentences are lacking the verb and often it is difficult the comprehension of the text.
In the INTRODUCTION
Major
- The introduction should be better organized because the author describes AQP7 (line 38) and then discuss about aquaglyceroporins without describing AQP7 belonging to the aquaglyceroporin group.
- In the introduction from the line 42, the author should change the sentence by adding that the also aquaglyceroporins (AQP7 and AQP9) have been investigated also in efferent ducts and epididymis in dog by examining their collaborative role hypothesizing their changed activities according to different stimuli (as a cause of cryptorchid condition) (see reference to add Squillacioti C. et al., Aquaporins are differentially regulated in canine cryptorchid efferent ductules and epididymis, Animals, 2021, 11, 1539).
- The author should describe better in the aim why decided to study AQP7 expression differently by other aquaglyceroporin. Moreover, several refences along the manuscript should be changed with others more specific.
Minor
- The sentence at page 1 (lines 35-36) lacks the verb.
- At the page 1, the References 1 and 2 should be changed with 1. Ishibashi K, Hara S, Kondo S (2009) Aquaporin water channels in mammals. Clin Exp Nephrol 13:107–117 and 2. Kreida S & Toernroth-Horsefield S (2015) Structural insights into aquaporin selectivity and regulation. Curr. Opin. Struct. Biol. 33:126-134
In the MATERIALS AND METHODS
Major
The author should better describe the group of camel bulls according the age in a specific Table (the animal group is homogeneous for the age?) indicating also the period rutting and non-rutting season during which the tissue samples were collected. The homogeneity of the group is important because other authors reported influence of age > 10 years on the Testicular Measurements and Semen Characteristics (see manuscript by 1. Al-Azhar, Influence of Seasons of the Year and Ages on the Testicular Measurements and Semen Characteristics of the Male Dromedary Camels, Journal of Agricultural Research V. (47) No. (1) June (2022) 89-98, 2. M.A. El-Harairy *, K.A. Attia, Effect of age, pubertal stage and season on testosterone concentration in male dromedary camel. Saudi Journal of Biological Sciences (2010) 17, 227–230.
The author evaluated some parameters of these animals prior to collect tissues samples?
- I would suggest the author to simplify the experimental sampling with the acronyms used (as reported in table 1.) because this organization often causes confusion. It could be better add the letter T or E near the words testis and epididymis respectively and indicate with the other letters the subsections of each tissue without repeat the first letter.
- Referring to the tissue’s samples, it should explain on the basis of what criteria, the author set the boundary of the individual segments. This is important especially in the case of in vitro culture, protein, and gene studies.
- Regarding the AQP7 antibody used, the author reported in the immunohistochemistry the image with negative control, but what about positive control ? (the datasheet of the ab 15123 is a polyclonal antibody that cross reacts with rat (not mouse monoclonal as reported in line 68 shows). The author should show the positive control.
Minor
- At line 56 substitute the singular verb was with “were”;
- Modify Table.1 with “Table 1.”
- The sentence (lines 57-59) lacks the verb.
- Substitute camel with “camels” at the line 66
- Add a full stop prior of A at the line 68.
- Add the verb in the sentence (lines 78-79)
- Add the code of the kit and the city and country for the reagent DNase kit (lines 81-82)
- Modify Table.2 with “Table 2” (line 91).
- In the legend of table 2 there is a mistake. Please correct.
RESULTS
Major
- I would suggest modifying the Figure 1 to be clearer. Moreover, the y axis should indicate the title: “AQP7 mRNA relative expression”
- Referring to the Table 3 how many slides were used for each sample for each animal? How many independent observers analyzed the slides?
- Referring to the Figures 2-4 (testis) they should reorganize in one single Figure in one complete page, reporting in a left column in high part the condition (rutting season) and on the right (non-rutting season). In this way it should abbreviate also the figure legend that actually is too repetitive. Some consideration also for figure 5-7, 8-10 and 11-12.
DISCUSSION
Major
The discussion should be better reorganized focusing on the results obtained. In particular the first part of discussion (focused on reproductive seasonality) could be associated with the results obtained relative to the AQP7 expression. Moreover, the results relative to the AQP7 expression at the level of basal cells in the body of epididymis (lines 320 -321) can be supported also by the findings of the paper by Squillacioti C. et al., Aquaporins Are Differentially Regulated in Canine Cryptorchid Efferent Ductules and Epididymis. Animals 2021, 11, 1539.
Author Response
Dear Dr. Nicolas Gai
Editor-in-Chief
Animals Journal
14/03/2023
Thank you for giving us the opportunity to submit a revised draft of my manuscript (animals-2217049) entitled “Morphological and Molecular Investigations of Aquaporin-7 (AQP-7) in Male Camelus Dromedarius Reproductive Organs” to Animals journal. I appreciate the time and effort that you and the reviewers have dedicated to providing your valuable feedback on the manuscript. I grateful to the reviewers for their insightful comments on the paper. I have been able to incorporate changes to reflect most of the suggestions provided by the reviewers. I have highlighted the changes within the manuscript.
Below is a point-by-point response to comments and concerns
Point 1: INTRODUCTION: The introduction should be better organized because the author describes AQP7 (line 38) and then discuss about aquaglyceroporins without describing AQP7 belonging to the aquaglyceroporin group.
Response 1: I agree with this comment. I rewrite it as : AQPs are grouped in 1) orthodox aquaporins (AQP0, AQP1, AQP2, AQP4, AQP5, AQP6 and AQP8) that are considered primary selective to water, 2) aquaglyceroporins (AQP3, AQP7, AQP9 and AQP10) that in addition to water, also transport glycerol and other small neutral solutes; and 3) subcellular or unorthodox aquaporins (AQP11 and AQP12) with distinct evolutionary pathway, localizing in intracellular membranes
-Ishibashi, K., Tanaka, Y., and Morishita, Y. (2014). The Role of Mammalian Superaquaporins inside the Cell. Biochim. Biophys. Acta 1840 (5), 1507–1512. doi:10.1016/j.bbagen.2013.10.039
Point 2: In the introduction from the line 42, the author should change the sentence by adding that the also aquaglyceroporins (AQP7 and AQP9) have been investigated also in efferent ducts and epididymis in dog by examining their collaborative role hypothesizing their changed activities according to different stimuli (as a cause of cryptorchid condition) (see reference to add Squillacioti C. et al., Aquaporins are differentially regulated in canine cryptorchid efferent ductules and epididymis, Animals, 2021, 11, 1539).
Response 2: I agree with this comment. I rewrite it as Aquaglyceroporins (AQP7 and AQP9) have been investigated also in efferent ducts and epididymis in dog by examining their collaborative role hypothesizing their changed activities according to different stimuli (as a cause of cryptorchid condition) instead of : Although extensive data exist on the physiological roles of aquaporin-facilitated water transport, until recently the biological significance of glycerol transport by the mammalian aquaglyceroporins has been unknown.
-Squillacioti C, Mirabella N, Liguori G, Germano G, Pelagalli A. Aquaporins Are Differentially Regulated in Canine Cryptorchid Efferent Ductules and Epididymis. Animals (Basel). 2021 May 25;11(6):1539. doi: 10.3390/ani11061539. PMID: 34070358; PMCID: PMC8227126.
Point 3: The sentence at page 1 (lines 35-36) lacks the verb.
Response 3: I agree with this comment. I rewrite it as Mammalian AQPs (AQP0-12) are grouped according to their primary structure [2] and permeability [3] into: 1) orthodox aquaporins (AQP0, AQP1, AQP2, AQP4, AQP5, AQP6 and AQP8) that are considered primary selective to water, 2) aquaglyceroporins (AQP3, AQP7, AQP9 and AQP10) that in addition to water, also transport glycerol and other small neutral solutes; and 3) subcellular or unorthodox aquaporins (AQP11 and AQP12) with distinct evolutionary pathway, localizing in intracellular membranes
Point 4: At the page 1, the References 1 and 2 should be changed with 1. Ishibashi K, Hara S, Kondo S (2009) Aquaporin water channels in mammals. Clin Exp Nephrol 13:107–117 and 2. Kreida S & Toernroth-Horsefield S (2015) Structural insights into aquaporin selectivity and regulation. Curr. Opin. Struct. Biol. 33:126-134
Response 4: I agree with this comment. References 1 and 2 changed
Point 5: MATERIALS AND METHODS : The author should better describe the group of camel bulls according the age in a specific Table (the animal group is homogeneous for the age?) indicating also the period rutting and non-rutting season during which the tissue samples were collected. The homogeneity of the group is important because other authors reported influence of age > 10 years on the Testicular Measurements and Semen Characteristics (see manuscript by 1. Al-Azhar, Influence of Seasons of the Year and Ages on the Testicular Measurements and Semen Characteristics of the Male Dromedary Camels, Journal of Agricultural Research V. (47) No. (1) June (2022) 89-98, 2. M.A. El-Harairy *, K.A. Attia, Effect of age, pubertal stage and season on testosterone concentration in male dromedary camel. Saudi Journal of Biological Sciences (2010) 17, 227–230.
Response 5: I agree with this comment but sometimes it must be taken into our consideration that these samples are taken from a slaughterhouse, and there is a limited number of adult males slaughtered weekly or monthly, so it was difficult to bias the age in this study, all males were healthy sexually mature and have mated previously
Point 6: Referring to the tissue’s samples, it should explain on the basis of what criteria, the author set the boundary of the individual segments. This is important especially in the case of in vitro culture, protein, and gene studies.
Response 6: the boundaries of the individual segments and tissue sampling were under the supervision of specialist in anatomy (Anatomy Department of College of Veterinary Medicine – King Faisal University according to these references:
- The Anatomy of the Domestic Animals:
- Author: Septimus Sisson
- Publisher: Philadelphia, London, W.B. Saunders Company, 1967.
- Color Atlas of Large Animal Applied Anatomy:
- Author: Hilary M. Clayton, Peter F. Flood
- Publisher: Mosby-Wolfe, SaintLouis, Missouri, 1996.
- Textbook of Veterinary Anatomy:
- Author: Dyce, Sack, and Wensing's
- Publisher: Philadelphia, London, W.B. Saunders Company, 2009.
- Textbook and Colour Atlas of Veterinary Anatomy of Domestic Animals:
- Author: Horst Erich Koenig
- Publisher: : Thieme Publishing Group, 2004
Point 7: Regarding the AQP7 antibody used, the author reported in the immunohistochemistry the image with negative control, but what about positive control ? (the datasheet of the ab 15123 is a polyclonal antibody that cross reacts with rat (not mouse monoclonal as reported in line 68 shows). The author should show the positive control.
Response 7: In this study the positive control was camel kidney. AQP7 antibody labels the brush border of the proximal tubule in the kidney
Point 8: At line 56 substitute the singular verb was with “were”;
Response 8: I agree with this comment. I rewrite it : All animal sampling procedures were performed according to the animal slaughter protocols and ethical guidelines of the Ministry of Municipal, Rural Affairs, and Housing, Saudi Arabia. The King Faisal University Ethics Committee approved the animal sampling (Ref. No. KFU-REC-2023-MAR-ETHICS718
Point 9: Modify Table.1 with “Table 1.”.
Response 9: I agree with this comment. I rewrite it Table 1.
Point 10: The sentence (lines 57-59) lacks the verb.
Response 10: I agree with this comment. I rewrite it as : The tissues samples as shown in Table 1 were collected
Point 11: Substitute camel with “camels” at the line 66
Response 11: I agree with this comment. I rewrite it camels
Point 12: Add a full stop prior of A at the line 68.
Response 12: changed
Point 13: Add the verb in the sentence (lines 78-79)
Response 13: : I agree with this comment .AQP-7 mRNA expression levels were evaluated using a semi-quantitative real- time RT-PCR analysis
Point 14: Add the code of the kit and the city and country for the reagent DNase kit (lines 81-82)
Response 13: I agree with this comment. I rewrite it a DNase I kit (Ambion, AM2222, Carlsbad, CA, USA).
Point 15: Modify Table.2 with “Table 2” (line 91).
Response 15: I agree with this comment. I rewrite it (Table 2.)
Point 16: In the legend of table 2 there is a mistake. Please correct.
Response 16: I agree with this comment. I rewrite it as : Real-time qPCR primers sequences targeting gene expression of AQP7.
Point 16: I would suggest modifying the Figure 1 to be clearer. Moreover, the y axis should indicate the title: “AQP7 mRNA relative expression”.
Response 16:. I add the title “AQP7 mRNA relative expression”on the y axis of Figure 1
Point 17: Referring to the Table 3 how many slides were used for each sample for each animal? How many independent observers analyzed the slides?
Response 17:
how many slides were used for each sample for each animal? The answer is 3/ sample
How many independent observers analyzed the slides? The answer is 4 observers / sample
Point 18: Referring to the Figures 2-4 (testis) they should reorganize in one single Figure in one complete page, reporting in a left column in high part the condition (rutting season) and on the right (non-rutting season). In this way it should abbreviate also the figure legend that actually is too repetitive. Some consideration also for figure 5-7, 8-10 and 11-12.
Response 18: Figures were changed as suggested
Point 18: DISCUSSION : the results relative to the AQP7 expression at the level of basal cells in the body of epididymis (lines 320 -321) can be supported also by the findings of the paper by Squillacioti C. et al., Aquaporins Are Differentially Regulated in Canine Cryptorchid Efferent Ductules and Epididymis. Animals 2021, 11, 1539.
Response 18: reference added, Our finding showed that, the highest gene expression of AQP-7 was in the body of epididymis compared to head and tail in both rutting and non-rutting males. In addition, the immune reactivity levels of anti-AQP-7 antibody was intense in the head, body and tail compared to the weak reactivity in of the same tissues of non-rutting males. The signal was strong in basal cell (stem cell) layer and sterocelia of the columnar cell of the head, body and tail of the epididymis. This was compatible with the finding of Squillacioti et al. that shows regional tissue distributions of AQP7, particularly at the level of the epithelium of efferent ductules and both the regions caput and cauda of the canine cryptorchid epididymis
Reviewer 2 Report
Comments about the manuscript:
“Morphological and Molecular Investigations of Aquaporin-7 (AQP-7) in Male Camelus Dromedarius Reproductive Organs”
Aquaporins (AQP) are involved in bidirectional transfers of water and small solutes across cell membranes. They are predentent in all tissues. However, the expression of AQP-7 has not yet been demonstrated in the reproductive tract of the camelid Camelus dromedarius. The study presented here concerns the immunohistochemical evidence of aquaporin-7 (AQP-7) in different parts of the male genital tract of Camelus dromedarius. This interesting study brings new elements to the knowledge of aquaporins. It can be published after some improvements to the manuscript. Here are a few remarks.
Title: write Camelus dromedarius using italics and a small first letter for the name of species.
Page1, line 9: give the scientific name of camel: "Camelus dromedarius" (italics).
Line 30: write Camelus dromedarius (italics with a small letter at the beginning of species).
Introduction: The introduction is short and needs to be developed, especially with regard to the distribution and explanation of the mechanisms of action of aquaporins.
Lines 40, 41: write “epididymis [8], and ductus deferens” instead of “Epididymis [8], and Ductus Deferens” (use lower cases).
Page 2, line 45: write “Camelus dromedarius” (italics) instead of “camelus dromedaries”
Page 3, line 58. Write "Table 1" instead of "Table .1": delete the period, and delete it for all other table calls throughout the text.
Page 3, line 66. Write “from 12 male camels during rutting” instead of “from 12 male camel during rutting” (there are several camels).
Page 3, lines 67-68. Write “for immunohistochemical examination A. mouse monoclonal” instead of “for immunohistochemical examination A mouse monoclonal” (add a period).
Page 3. 2.2. Immunohistochemistry Paraffin Protocol (IHC-P) for AQP-7 detection.
The section on immunohistichemistry needs to be completed. Summarize the technique. Duration of incubation of antibodies? What is the second antibody? Is it biotinylated? It seems that ATM was used as a chromogen. Please clarify. Also specify how the negative checks were carried out.
Page 3. 2.3. RNA Isolation and Semi-Quantitative Real Time RT-PCR Analysis
specify the method: it is not sufficient to indicate the reference of the kit. This part also needs to be completed.
Page 4, line 112. Write “rete testis” instead of “reti testis”.
Page 5, line 136. Write “Figures 2-4.” Instead of “Figure. 2- 4.” (delete period)
Figures 2 to 12.
How was this control performed (see my remarks in material and methods)
Add a scale bar on the images (the scale bar varies with the enlargement of the edited text, the magnification indication becomes false).
Page 13, lines 293-294. Write “the finding of Marai et al. [12] in that” instead of “the finding of [12] (Marai et al. 2009) in that”.
Page 14, lines 324-325. Write “This may indicate that AQP-7 may also” instead of “This may indicate that, AQP-7 may also” (delete comma).
References
Check the references to see if they are presented according to the standards of the journal (some references are capitalized and the beginning of each word, others are written with a lower case).
Author Response
Dear Dr. Nicolas Gai
Editor-in-Chief
Animals Journal
14/03/2023
Thank you for giving us the opportunity to submit a revised draft of my manuscript (animals-2217049) entitled “Morphological and Molecular Investigations of Aquaporin-7 (AQP-7) in Male Camelus Dromedarius Reproductive Organs” to Animals journal. I appreciate the time and effort that you and the reviewers have dedicated to providing your valuable feedback on the manuscript. I grateful to the reviewers for their insightful comments on the paper. I have been able to incorporate changes to reflect most of the suggestions provided by the reviewers. I have highlighted the changes within the manuscript.
Below is a point-by-point response to comments and concerns
Point 1: Title: write Camelus dromedarius using italics and a small first letter for the name of species.
Response 1: I agree with this comment. I rewrite it italics
Point 2: Page1, line 9: give the scientific name of camel: "Camelus dromedarius" (italics).
Response 2: I agree with this comment. I rewrite it as in male Camelus dromedarius testes
Point 3: Line 30: write Camelus dromedarius (italics with a small letter at the beginning of species).
Response 3: I agree with this comment. I rewrite it italics
Point 4: Lines 40, 41: write “epididymis [8], and ductus deferens” instead of “Epididymis [8], and Ductus Deferens” (use lower cases).
Response 4: I agree with this comment. I rewrite it in lower cases
Point 5: Page 2, line 45: write “Camelus dromedarius” (italics) instead of “camelus dromedaries”
Response 5: I agree with this comment. I rewrite it italics
Point 6: Page 3, line 58. Write "Table 1" instead of "Table .1": delete the period, and delete it for all other table calls throughout the text.
Response 6: I agree with this comment. I rewrite it "Table 1" and I have delete “during rutting and non-rutting season”
Point 7: Page 3, line 66. Write “from 12 male camels during rutting” instead of “from 12 male camel during rutting” (there are several camels).
Response 7: I agree with this comment. I rewrite it from 12 male camels during rutting
Point 8: Page 3, lines 67-68. Write “for immunohistochemical examination A. mouse monoclonal” instead of “for immunohistochemical examination A mouse monoclonal” (add a period).
Response 8: I agree with this comment. I rewrite it “for immunohistochemical examination A. mouse monoclonal”
Point 9: Page 3. 2.2. Immunohistochemistry Paraffin Protocol (IHC-P) for AQP-7 detection.
The section on immunohistichemistry needs to be completed. Summarize the technique. Duration of incubation of antibodies? What is the second antibody? Is it biotinylated? It seems that ATM was used as a chromogen. Please clarify.
Response 9: I agree with this comment. I rewrite it as : Approximately one centimeter of each tissue mentioned in Table 1. was collected from 12 male camels during rutting and non-rutting seasons. Then fixed overnight in 4% paraformaldehyde. All the fixed samples were processed for immunohistochemical examination , according to the following steps: dehydration, clearing, infiltration, embedding and sectioning. Then tissue sections were deparaffinized by passing the thermo-scientific super frosted plus charged slides in xylene; 2 changes, 5 min each and followed by hydration of the sections by dipping them for 30s through degraded alcohol (100%, 100%, 95%, 80%, 70% and washed the slides 2 x 5 min in tris-buffered saline (1X TBS) plus 0.025% triton X100 with gentle agitation. Then blocking in 10% normal serum with 1% bovine serum albumin (BSA) in (1X TBS) for 2 hours at room temperature. Mouse and rabbit specific HRP/DAB (ABC) detection IHC kit (ab64264) was used to detect anti-AQP-7 antibody (Abcam, Ab15123) as following: enough drops of hydrogen peroxide block was added to cover the sections and incubated for 10 min at room temperature, then washed 2 times in 1X TBS buffer, protein block was applied on sections and incubated for 10 min at room temperature, then washed 2 times in 1X TBS buffer, the primary mouse anti-AQP-7 antibody (Abcam, Ab15123) diluted in 1X TBS with 1% BSA and applied on sections and incubated overnight at 4°C.
In the next Day sections were rinsed 4 times in 1X TBS 0.025% triton with gentle agitation and biotinylated goat anti-polyvalant enzyme-conjugated secondary antibody was applied on the sections and incubated for 1 hour at room temperature and washed 4 times in 1X TBS 0.025% triton. Streptavidin peroxidase was applied to cover the sections and incubated for 10 min at room temperature and rinsed 4 times in 1X TBS 0.025% triton., 30μl DAB substrate solution was applied to cover the sections until the desired color intensity is reached and then washed with PBS 3 times, 2 min each. Then sections were dehydrated by dipping them in graded alcohol (70%, 95%, 100%, 100%) just for seconds and clearing them be using xylene; 2 changes, 5 min each. Finally, the sectioned tissues were mounted by using DPX with cover slides 22x40mm and the staining was observed by Leica ICC50 W light microscopy under 10x and 40x magnification powers, Wi-Fi-capable digital camera detector and Leica AirLab App software. Reaction reactivity was taken using image processing and analyzed using an imageJ 1.52a analyzer (Wayne Rasband, National Institute of Health, USA, http://imagej.nih.gov/ij).
Point 10: Page 3. 2.3. RNA Isolation and Semi-Quantitative Real Time RT-PCR Analysis
specify the method: it is not sufficient to indicate the reference of the kit. This part also needs to be completed.
Response 10: I agree with this comment. I rewrite it as : AQP-7 mRNA expression levels evaluated using a semi-quantitative real- time RT-PCR analysis. Tissues were homogenized by Bead Ruptor (24 Bead Mill Homogenizer, OMNI, USA) and total RNA was extracted using the PureZOL TM RNA isolation method (BIO-RAD, Catalog #732-6890, Hercules, CA, USA) by adding 1 ml of PureZOL for every 100 mg of tissue in 2ml sized tube for disruption and homogenization by Bead Ruptor (24 Bead Mill Homogenizer OMNI, USA), then 0.2 ml of chloroform was added per 1 ml of PureZOL and incubate for 5 minutes at room temperature and centrifuge at 12,000 x g for 15 minutes at 4°C. The aqueous phase was immediately transferred to a new RNase-free tube. 0.5 ml of isopropyl alcohol was used per 1 ml of PureZOL and then incubate at room temperature for 5 minutes then centrifuged at 12,000 x g for 10 minutes at 4°C. RNA pellet, was washed by adding 1 ml of 75% ethanol and then centrifugeed at 7,500 x g for 5 minutes at 4°C. RNA pellet was allowed to air-dry for about 5 minutes and then resuspend in 100 µl of RNasefree water. DNA was removed using a DNase I kit (Ambion), and the RNA samples were checked for their concentration and purity (260:280 nm absorbency) using a Synergy™ Mx Monochromator-Based Multi-Mode Microplate Reader (Bio-Tek, USA). RNA (2 μg) was reverse transcribed to cDNA in a reaction mixture using an iScriptc DNA synthesis kit (BIO-RAD, Catalog #170-8890, Hercules, CA, USA).
Point 11: Page 4, line 112. Write “rete testis” instead of “reti testis
Response 11: I agree with this comment. I rewrite it “rete testis” instead of “reti testis
Point 12: Page 5, line 136. Write “Figures 2-4.” Instead of “Figure. 2- 4.”
Response 12: I agree with this comment. I rewrite it Figures 2-4.” Instead of “Figure. 2- 4.”
Point 13: Add a scale bar on the images (the scale bar varies with the enlargement of the edited text, the magnification indication becomes false).
Response 13: scale bar was added
Point 14: Page 13, lines 293-294. Write “the finding of Marai et al. [12] in that” instead of “the finding of [12] (Marai et al. 2009) in that”
Response 13: I agree with this comment. I rewrite it the finding of Marai et al. [12]
Point 15: Page 14, lines 324-325. Write “This may indicate that AQP-7 may also” instead of “This may indicate that, AQP-7 may also” (delete comma).
Response 15: I agree with this comment. I rewrite it “This may indicate that AQP-7 may also
Point 16: References
Check the references to see if they are presented according to the standards of the journal (some references are capitalized and the beginning of each word, others are written with a lower case).
Response 16: I agree with this comment. References were checked

Round 2
Reviewer 1 Report
Dear author,
the manuscript has been revised according to the reviewers comments and now its quality and the presentation of results is improved.
the manuscript in this revised form is suitable for publication.